# Left-Behind Children and Risk of Unintentional Injury in Rural China—A Cross-Sectional Survey

**DOI:** 10.3390/ijerph16030403

**Published:** 2019-01-31

**Authors:** Sha Ma, Minmin Jiang, Feng Wang, Jingjing Lu, Lu Li, Therese Hesketh

**Affiliations:** 1The Institute of Social and Family Medicine, Zhejiang University School of Medicine, 866 Yuhangtang Lu, Hangzhou 310012, China; masha.1987@163.com (S.M.); jiangm@zjnu.edu.cn (M.J.); wangfeng1990@zju.ed.cn (F.W.); jingjinglu@zju.edu.cn (J.L.); lilu@zju.edu.cn (L.L.); 2The Centre for Global Health, Zhejiang University School of Medicine, 866 Yuhangtang Lu, Hangzhou 310012, China; 3The UCL Institute for Global Health, 30 Guilford St, London WC1N1EH, UK

**Keywords:** left-behind children, environment, unintentional injury, China

## Abstract

Unintentional injury is the leading cause of childhood death and disability in many countries worldwide. This study aimed to quantify rates and risk factors for childhood unintentional injury in areas of rural China, where many children are left behind by migrant worker parents. We administered a questionnaire to children aged 9 to 15, in 56 schools in five counties in Zhejiang and Guizhou provinces. Of the 3791 respondents, 44% lived with both parents, 23% with one parent, and 33% with neither. Around half the children (47.9%) had suffered at least one unintentional injury in the past year, with burns (26%), animal bites (20%) and mechanical injury (18%) the most common. Left-behind children had no increased risk of unintentional injury, but children living in poorer Guizhou (*p* = 0.001), of divorced parents (*p* = 0.02), and less well-educated mothers (*p* = 0.02) were associated with higher risk. Virtual absence of personal level risk factors highlights the importance of addressing environmental risk to reduce childhood injury. The findings have informed a community-based intervention to reduce injury risk through raising awareness of environmental hazards, and through removal of specific hazards. Importantly, the Chinese government should ensure that known effective interventions are subject to legislation and enforcement.

## 1. Introduction

Children are especially vulnerable to unintentional injury, and it is the leading cause of death and disability in children in many low- and middle-income countries (LMIC) [1]. China has a particularly high burden of disease from unintentional injury in children. In 2010 it was the leading cause of death and disability in children under 18, with 86,000 children dying as a result of unintentional injury, accounting for 26% of all child deaths [2]. The commonest causes were drowning, road traffic injury, mechanical injury (mainly falls), poisoning and burns. The non-fatal incidence of unintentional injury in China is impossible to estimate on the current evidence because of widely varying definitions of injury, and the range of locations and age groups investigated in research studies [3,4,5,6,7]. What is clear is that unintentional injury not only has devastating effects for children and families but is also associated with considerable healthcare and social costs [8].

Studies show that the risk of unintentional injury in children is affected by three major factors: (1) Children’s physical and cognitive abilities, mainly determined by age, (2) environmental or neighborhood factors, and (3) social factors, including poor care and supervision by adult caregivers [9]. Relevant environmental factors depend on the context. In urban areas, most moderate or severe injuries are traffic-related, followed by falls and poisoning resulting from ingestion of household chemicals or pharmaceuticals. In rural areas, especially in low- and middle-income countries, injuries are related mainly to farming activities, poisoning (usually from pesticides), falls and drowning [9,10,11]. Social factors are also crucial. A systematic review of the global literature has shown an association between unintentional injury and social deprivation, which was primarily defined in economic terms [12]. The association with social deprivation is seen as related to inadequate care and supervision.

Such lack of adequate care and supervision is a particular risk in rural areas of many LMICs, where large numbers of children are left behind by parents who are forced to migrate from rural areas for work, in order to improve their living standards [13]. Children are most often left in the care of family members, usually grandparents [14]. In many settings this leads to concerns about the quality of care and supervision of these so-called Left-Behind Children (LBC) [15]. Total numbers of such left-behind children globally are unknown. But it is estimated that there are three to six million left-behind children in the Philippines, at least one million in Indonesia, and half a million in Thailand [16]. China greatly exceeds all of these: Massive rural-urban migration has led to around 60 million left-behind children, accounting for 38% of all Chinese rural children and 22% of all children. An estimated 29 million live with neither parent, and over 2 million live alone [17].

Two studies have shown that left-behind children in China are especially vulnerable to all types of unintentional injury because of poor care and supervision [6,18]. In addition, a number of extreme events, including fatal injuries, drowning and suicide, using pesticides, among rural Chinese children have been widely reported by the media and have drawn attention to the consequences of the phenomenon of the large numbers of left-behind children in rural areas with lack of appropriate supervision and exposure to environmental hazards. This led the Chinese government in 2013 to call on local government to prioritize improvements in the care of children left behind by migrant parents [19]. Partly in response to this call, and in recognition of the vulnerability of left-behind children across a range of parameters, we started to work with local governments in two provinces, Zhejiang and Guizhou, to explore the development of an intervention to provide support and care to children in poor rural areas, with a focus on left-behind children. The intervention was planned for five migrant-sending counties in the two provinces. Zhejiang is a wealthy eastern province ranked fifth in GDP per capita of all 31 provinces. In contrast Guizhou, in south-west China, ranks 29. As part of the baseline survey assessment for the project, we conducted a survey of rural children. Local authorities had expressed particular concern about injury risk for children and wanted to know whether being left behind put children at greater risk, so specific questions about this were included.

So, the aims of this study were to: (1) Measure the incidence of unintentional injury in left-behind children compared with non-left-behind children, (2) identify the risk factors associated with children’s unintentional injury, and (3) inform the proposed intervention and local policy measures.

## 2. Materials and Methods

### 2.1. Participants

The cross-sectional survey was carried-out from November 2015 to May 2016 among children aged nine to fifteen years, recruited at 56 primary and middle schools in migrant-sending areas of the three counties in Zhejiang province, and the two counties in Guizhou province. In all five counties local government had expressed an interest in collaborating on an intervention for vulnerable children. The baseline survey was conducted in the townships where the intervention was planned, with all schools in the intervention townships involved. Head teachers in all schools were approached and all agreed to participate. Across the 56 schools, one class in each year group from age 9 to 15 (grade 3 to grade 6) was randomly selected for inclusion in the study. The cut-off age of nine was chosen because literacy in Chinese (being a character-based language) cannot be assumed until around the age of nine years.

Pairs of researchers (one pair for each province) conducted the survey, spending one day at each school. The class teachers introduced the researchers and explained the study. Then the self-completion questionnaires were administered by the researchers, who provided an explanation about how to complete the questionnaire, how it was anonymous and confidential and not compulsory. Written consent was included on the first page of the questionnaire, and all respondents were required to sign it. This page was then detached and stored separately to ensure anonymity. The researchers stayed in the classroom to deal with queries. For a small number of the 9–10 year-olds, the researchers administered the questionnaire face-to-face because of literacy limitations in some children in this age group.

### 2.2. Measures

The questionnaire was designed specifically for this study. The socio-demographic section included a key question, who the child currently lived with, length of absence of parent(s) if applicable, and who was regarded as the primary carer, that is, the adult who takes most of the parenting role. Left-Behind Children were defined according to the standard definition, as children who have lived apart from one or both migrant parents for at least six months. Children are unlikely to be aware of household income, so economic status was assessed with a question about perceived wealth: What do you think about your family’s economic situation in comparison with that of your classmates: better, similar, worse, or don’t know.

We asked about unintentional injury experienced in the previous 12 months. Unintentional injury was divided into mechanical (including falls, abrasions, bruises, lacerations, fracture and sprain), animal bites (dog, cat, snake), burns and scalds, near drowning, and road traffic injury. Children were asked to briefly describe the injury and the sequelae. 

The questionnaire was piloted across the age range in Zhejiang and Guizhou and amendments were made accordingly. Permission was obtained from the schools. Informed consent was obtained from the children at the time of the survey and from primary carers through the headteachers. Ethical approvals were obtained from University College London Research Ethics Committee and the Ethics Committee of Zhejiang University (Project Number ZJU 20141202).

### 2.3. Analysis

Chi-squared tests were used to determine the association between children living with both parents, one parent or with no parents, and unintentional injuries overall, as well as by type. Logistic regression was used to control for potential confounders. Injury risk was presented using Odds Ratios and 95% confidence intervals.

## 3. Results

### 3.1. Socio-Demographic Characteristics (Table 1)

We distributed a total of 3992 questionnaires, of which 3876 (97.1%) were returned. Of these, 85 were excluded from analysis because of failure to complete key variables, so the total response rate was 94.9% (*n* = 3791). The age range of participants was 9 to 15 years with a mean of 13.5. Overall 59% of the children were from Guizhou, because average class size there was considerably larger than in Zhejiang. There were 1254 (33.2%) children living with neither parent, 861 (22.7%) living with one parent, and 1676 (44.2%) living with both parents, with more left-behind children in Guizhou (57.9%) than Zhejiang (49.7%). There were no children living entirely alone, that is without the care of an adult. There were a few instances (less than 20 overall) where 14 or 15 year olds lived next door to grandparents, were supervised by them, had all meals with them, but slept in the parental home.

The family economic situation was perceived as worse than others in the community by children living with both parents (50%) compared with 28% of children left behind by both parents, and 22% left behind by one parent (*p* < 0.001). Overall 12% of children had divorced parents. Mothers with a higher education were more likely to migrate, while level of education was not an influence for fathers. The primary carer was stated as the grandparent by 1802 (49.5%), including 30% living with one parent, and 22% living both parents (Table 1).

Overall children left behind by one or both parents were significantly more likely to be from Guizhou, be female, older, from wealthier households, have better educated mothers, and have parents who were more likely to be divorced.

### 3.2. The Incidence of Unintentional Injuries (Table 2)

Of the total, 1811 (47.9%) participants had suffered at least one type of unintentional injury in the 12 months before the survey (Table 2). Burns and scalds were the most common injury reported (26%), followed by animal bite (20.4%) mostly by dogs, and mechanical injury (17.9%) of a range of types, with falls most commonly cited. Near drowning accounted for 6%, that is the child, unable to swim, got into difficulty in water. Road traffic injury, both bike and pedestrian injuries, accounted for 7.7%. The overall incidence of unintentional injury was significantly, but only slightly, higher in left-behind children than children living with one parent, and non-left-behind children (*p* = 0.04). However, when categorized by type of injury there were no significant differences between the three groups.

### 3.3. Association with Unintentional Injuries (Table 3)

Before adjustment a number of significant associations between socio-demographic characteristics and unintentional injuries in the past 12 months emerged (Table 3): Children living in Guizhou compared with Zhejiang (OR 1.25, 1.11–1.47, *p* < 0.01) children left behind by one or both parents (OR 1.15, 1.01–1.32, *p* = 0.04), children with less well-educated parents: father (OR 1.19, 1.04–1.36, *p* = 0.01), mother (OR 1.29, 1.12–1.47, *p* = 0.01), children of divorced parents (OR 1.39, 1.47–1.70, *p* < 0.001) and children whose primary carer is a grandparent (OR 1.16,1.02–1.33, *p* = 0.02). However, after adjustment for residence, age, parental education, parental marriage, and primary caregiver, children living in Guizhou, (OR 1.28, 1.1–1.45, *p* = 0.01, of divorced parents (OR 1.29, 1.04–1.59, *p* = 0.02) and with mothers with a low education level (OR 1.22, 1.03–1.42, *p* = 0.02) were more likely to have experienced an injury.

Left behind status was not significant after adjustment. After disaggregation by injury type, we also found there were no significant associations between potential determinants and injury type between left-behind and non-left-behind children, after adjustment for residence (Zhejiang/Guizhou) parental education, and grandparental care.

## 4. Discussion

In this study around half of all children had suffered unintentional injuries in the previous year. Although some were relatively minor, the fact that they were recalled up to a year later suggests they did have an impact. Younger children, especially the under-fives, are regarded as at higher risk in almost all populations [4], with the age group we studied at lower risk, highlighting the high childhood disease burden from unintentional injury in rural China. The incidence we found is marginally higher compared with the few other studies [3,4,5,6,7], although direct comparison is hampered by the use of different measures, age ranges and locations. However, our study is strengthened by its size, across 56 schools in two provinces and five counties of varying geography and socio-economic conditions.

Our findings also challenge assumptions, widely publicized in the Chinese media and in some previous research [6,18], that LBC are especially vulnerable to all types of unintentional injury because of poor care and supervision. In our study left behind status of the child was not a predictor of unintentional injury across all injury types. After adjustment for confounders we found the only significant associations to be: residence in poorer Guizhou, low educational level of the mother, and parental divorce. However, it should also be noted that the odds ratios for all of these risk factors are low, and their actual importance therefore questionable. One other study from a county in central China, found parental divorce, to have the strongest association with unintended injury [6].

There are a number of possible explanations for our finding that LBC are not more vulnerable to unintentional injury, despite probable lower levels of supervision. Firstly, many injuries occur with or without supervision, so the level of supervision *per se* may not be relevant. The exception to this is probably near drowning, which we were especially interested in because of two cases of actual drowning in our study areas in the previous year. Indeed, the relatively high number of cases of near drowning, almost as many as road traffic injuries, is noteworthy. In the long hot summer holidays, children in these rural areas inevitably play around water, mainly rivers, with all the obvious risks for non-swimmers. Only a small minority of Chinese can swim with almost none in most rural areas [20]. Further follow-up investigation with local authorities determined that children living near rivers were more likely to report near drowning, emphasizing the environmental risk. Secondly, we found no evidence that grandparents, the major carers of left-behind children, are inferior to parents in caring for and supervising children. In fact, they may provide better protection than parents. In our study it was notable that 23% of children living with both parents described a grandparent as primary carer, so there is clearly heterogeneity in child care responsibility across all three groups. This phenomenon of more protective grandparental care was found in a study from the United States, which showed that injuries requiring medical care were less common in young children cared for by grandparents [21]. Thirdly, in small communities, such as those in our study, where residents know each other well, children who are left behind are usually known to neighbors and friends, who form an informal care network for such children, especially those in the care of frail grandparents. We have observed this for ourselves while working in these rural areas.

Our study has limitations. First, child self-report for events which are common in childhood over a 12-month period may not be accurate, but this is likely to lead to underestimates rather than overestimates, and there is unlikely to be a difference here between LBC and non-LBC. Contrary to our previous assumption we found that the recall of younger children was good and some described the trauma of the event to the researchers in some detail. Second, we did not ask about poisoning. Shortly before we conducted this survey there was a fatal poisoning, involving four children from one family close to our Guizhou project site. The local authorities requested that we did not ask children about poisoning because it had become a very sensitive issue. Thirdly, while we asked children to describe their injury, only 229 (12.7%) did, so we could not assess the overall severity of injuries. Of those who did respond to the request, nearly all had been taken to a healthcare facility, but this is clearly a very biased sample. Comparable studies have not attempted to assess severity. Fourthly, we conducted this research in areas where there was already local authority interest in vulnerable children (sufficient to want to be involved in an intervention), so there could already be increased awareness and a sense of community responsibility for left-behind children, thus perhaps reducing their exposure to injury.

Our study has clear policy implications. Our failure to identify individual level risk factors highlights the importance of environmental factors and the necessity of addressing these in order to reduce the burden of disease from childhood injury. Prevention here involves a two-pronged approach: (1) Reduction in environmental risk factors, and (2) improving awareness of the dangers of unintentional injury. It also involves both local approaches and governmental legislative approaches.

Given that this research was conducted to inform local interventions, we were able to immediately benefit from the findings. A WHO report has described the way in which community-based interventions have used relevant information on local patterns of injury and their causes and reduced the rates of injuries in many countries [9]. Prevention of injuries can be achieved through environmental modifications, applying and/or reinforcing regulatory measures, and, overall, changing unsafe behaviors through education.

We started to roll-out the community-based intervention across all five counties from March 2017. It consisted of “children’s clubs” in community halls or on school premises, which provided a place for play, educational and creative activities, as well as a place of safety. This was especially important in reducing environmental risk at week-ends and in the long summer vacation when the more serious injuries such as road traffic accidents and incidents of drowning are known to be most common. Secondly, we included specific education sessions about how to avoid injury. Parents and carers are invited to be involved in these sessions. We drew partly on the model described by Cao et al. in a county in south-west China [7]. In areas where there were rivers, ponds and lakes, there was discussion with local authorities about covering these water hazards but playing in water in the summer is a pleasure that, it was felt by some officials, should not be denied to children. So local decisions were made, with some townships and villages choosing to cover water, while others focused on raising awareness about the water depth and dangers of local water hazards, together with appropriate clear signage. Unfortunately, there is no suitable setting yet to safely teach children how to swim in any of the five counties. We are currently still analyzing the data. Preliminary results suggest while knowledge about injury has improved actual incidence of injury remains unchanged, but it is too early to draw firm conclusions.

At the governmental legislative level, a number of measures are necessary. There is now a solid body of evidence for measures which are effective in addressing environmental risks which cause unintentional injury in children. Much of this is based on legal requirements and prohibitions, which it is known can modify behaviors and reduce the risk of injury [22]. Evidence shows that legislation can substantially reduce child injuries caused by road traffic, drowning, burns, falls, and poisoning [20]. Examples include removing or covering water hazards, (as noted above) use of seat belts in vehicles, and child resistant packaging of poisons and medicines. However, the Chinese legislative framework in this area remains weak. In 2013 Li et al. analyzed the legislation coverage in China for 27 known effective interventions against injury-related child mortality [23]. Only seven were covered by legislative documentation of the State Council. Importantly, many of the legislative documents failed to assign responsibility for implementation to government departments, so enforcement is a challenge.

## 5. Conclusions

The incidence of unintentional injury can be reduced through known effective interventions. The Chinese government needs to ensure that these interventions are covered by national laws. Local authorities must be made aware of the need to enforce laws and regulations and must take measures to provide a safer environment for children. 

## Figures and Tables

**Table 1 ijerph-16-00403-t001:** Social-demographic characteristics of the participants.

Variables		Children Left-Behind by Both Parents		Children Left-Behind by One Parent		Living with Both Parents		Total		*x* ^2^	*p*
		*n*	%	*n*	%	*n*	%	*n*	%		
Province	Zhejiang	430	27.7	388	25.0	735	47.3	1553	41.0	34.87	0.001
	Guizhou	824	36.8	473	21.1	941	42.0	2238	59.0		
Gender	Male	581	32.4	376	21.0	835	46.6	1792	47.7	9.08	0.01
	Female	651	33.2	481	24.5	831	42.3	1963	52.3		
Age in years	9 to 11	375	33.5	307	27.4	438	39.1	1120	30.5	25.24	<0.0001
	12 to 15	826	32.3	534	20.9	1197	46.8	2557	69.5		
Perceived Wealth	Low	595	28.4	363	21.5	520	50.1	1478	39.1	87.23	<0.0001
	High	652	40.3	495	24.6	1151	35.2	2298	60.9		
Father’s education	Low	830	32.7	574	22.6	1135	44.7	2539	67.3	0.7	0.70
	High	417	33.7	284	23.0	535	43.3	1236	32.7		
Mother’s education	Low	785	31.5	549	22.0	1159	46.5	2493	66.1	15.82	<0.0001
	High	460	36.1	309	24.2	507	39.7	1276	33.9		
Parental divorce	Yes	226	50.3	90	20.0	133	29.6	449	12.0	72.9	<0.0001
	No	1009	30.6	762	23.1	1524	46.3	3295	88.0		
Primary carer	Parent	0	0.0	562	30.5	1279	69.5	1841	50.5		
	Grandparent	1118	62.0	295	16.4	389	21.6	1802	49.5		

Notes: High parental education was defined as high school or above.

**Table 2 ijerph-16-00403-t002:** Comparison of the specific unintentional injuries among three groups of children.

Unintentional Injury	Total		Left-Behind Children No Parents		Left-Behind Children One Parent		Non-Left-Behind Children		*x* ^2^	*p*
	*n*	%	*n*	%	*n*	%	*n*	%		
**All**	1811	47.9	645	51.4	406	47.2	760	45.3	4.33	0.04
**Mechanical injury**	675	17.9	238	19.0	151	17.6	286	17.1	1.85	0.4
**Animal bite**	772	20.4	276	22.1	167	19.4	329	19.7	3.2	0.21
**Burn**	991	26.2	337	27.0	230	26.7	424	25.4	1.11	0.57
**Near drowning**	226	6.0	83	6.7	53	6.2	90	5.4	2.11	0.35
**Traffic injury**	293	7.7	107	8.6	66	7.7	120	7.2	1.91	0.39

**Table 3 ijerph-16-00403-t003:** Association between sample characteristics and all unintentional injuries in participants.

Variables	The Number of Children with Unintentional Injuries		Crude OR (95% CI)	*p*	Adjusted OR (95% CI)	*p*
	Yes	No				
**Province**						
**Guizhou**	1120	1111	1.28 (1.11–1.47)	<0.01	1.25 (1.1–1.45)	<0.01
**Zhejiang**	691	861	Reference		Reference	
**Children’s type**						
**LBC**	628	622	1.15 (1.01–1.32)	0.04	0.94 (0.77–1.14)	0.51
**Non LBC**	1183	1350	Reference		Reference	
**Gender**						
**Male**	843	944	0.96 (0.85–1.09)	0.54	0.86 (0.73–1.03)	0.10
**Female**	944	1016	Reference		Reference	
**Age**						
**6 to 11**	514	603	0.92 (0.8–1.05)	0.22	0.83 (0.69–1.01)	0.06
**12 to 15**	1230	1322	Reference		Reference	
**Wealth**						
**High**	1105	1192	1.02 (0.89–1.16)	0.79	1.06 (0.92–1.23)	0.41
**Low**	704	773	Reference		Reference	
**Father’s education**						
**Low**	1179	1358	1.19 (1.04–1.36)	0.01	0.9 (0.76–1.07)	0.24
**High**	629	607	Reference		Reference	
**Mother’s education**						
**Low**	1139	1352	1.29 (1.12–1.47)	<0.01	1.22 (1.03–1.42)	0.02
**High**	665	611	Reference		Reference	
**Parental divorce**						
**Yes**	248	201	1.39 (1.14–1.7)	<0.01	1.29 (1.04–1.59)	0.02
**No**	1547	1746	Reference		Reference	
**Primary caregiver**						
**Grandparent(s)**	892	909	1.16 (1.02–1.33)	0.02	1.22 (0.72–1.05)	0.14
**Parent**	840	1000	Reference		Reference	

Adjusted for residence, sex, age, wealth, parental education, parental marriage, and primary caregiver.

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
