# Peer review of "Left-Behind Children and Risk of Unintentional Injury in Rural China—A Cross-Sectional Survey"

_ijerph, 2019, doi:10.3390/ijerph16030403_

Round 1
Reviewer 1 Report
Overall this is an acceptable article, but I have a few reservations. Firstly if the children had to sign their names on "the first page of the questionnaire" (line 97) the instrument can hardly be anonymous. I would hope that the page was detached and stored securely separately, but this is not stated. A more serious concern is the lack of information on response or refusal rate. The participating class sizes must be known more or less: a certain amount of churn could be present. This makes it impossible to judge the adequacy of the sample. There may be reasons why no totals were given - the research was after all in China which no doubt has its own limitations - but I see this as an important issue.
It would have perhaps strengthened the paper if an age-specific analysis had been presented. Children under 12 years old are different to older children: the information elicited from them could be less reliable and they may have different experiences of injury or recall of when they were injured.
Mention is made in the Introduction of the possibility that some left-behind children live alone (line 62). Were there any such in the sample? In Table 1 the number of LBC whose primary carer was a grandparent is about 11% short of their total number in the sample compared to the shortfall in the children living with at least one parent (about 0.5%). Failure to answer this question is surely unlikely. Some explication would help.
Author Response
We very much appreciate the reviewer’s time and trouble in making these comments. Our responses are in bold below.
Overall this is an acceptable article, but I have a few reservations. Firstly if the children had to sign their names on "the first page of the questionnaire" (line 97) the instrument can hardly be anonymous. I would hope that the page was detached and stored securely separately, but this is not stated.
Yes, of course this was the case – and we have clarified it in the text (Line 96).
A more serious concern is the lack of information on response or refusal rate. The participating class sizes must be known more or less: a certain amount of churn could be present. This makes it impossible to judge the adequacy of the sample. There may be reasons why no totals were given - the research was after all in China which no doubt has its own limitations - but I see this as an important issue.
Thank you for pointing-out our omission in this regard. We do of course have these data; since this questionnaire was administered in the classroom setting the response rates were very high. We distributed 3992 questionnaires, of which 3876 (97.1%) were returned. Of these 85 were excluded from analysis because of failure to complete key variables, so the total response rate was 94.9% (n=3791). We have added this to the text on Lines 125 to 127.
It would have perhaps strengthened the paper if an age-specific analysis had been presented. Children under 12 years old are different to older children: the information elicited from them could be less reliable and they may have different experiences of injury or recall of when they were injured.
We did analyse by year group initially, but found no overall differences in injury by age in the different age groups. Given the variation in maturity and personality of children in the two age groups, we were advised that dividing into two groups (9-11 and 12-15, or late primary and middle school), and ensuring large sample sizes when categorised by left behind status, was a more desirable approach. In terms of recall, our experience in talking with these children was that younger children seemed to demonstrate very good recall, describing the trauma of the event in ways that older children did not, or at least seemed to accept. See Line 222
Mention is made in the Introduction of the possibility that some left-behind children live alone (line 62). Were there any such in the sample? In Table 1 the number of LBC whose primary carer was a grandparent is about 11% short of their total number in the sample compared to the shortfall in the children living with at least one parent (about 0.5%). Failure to answer this question is surely unlikely. Some explication would help.
There were no children living entirely alone, that is without the care of an adult. In general these children who live alone are 16-17 year olds, who were not included in our study. There were a few instances (less than 20 overall) where 14 or 15 year olds lived next door to grandparents, were supervised by them, had all meals with them, but slept in the parental home. But we did not regard these as “living alone”. This note has been added on Lines 137-140.
Regarding the role of grandparents this refers partly to the role that grandparents play in children’s lives, irrespective of left behind status, and hence the numbers don’t add-up in the way that the reviewer assumes. As in much of China, children often view their grandparents as primary carers, even if living with both their parents, since both parents work long hours and parental duties are carried-out by grandparents. As is shown 21.6% of children living with both parents state that their grandparents are primary carers, compared with 16.4% of children living with one parent.
Reviewer 2 Report
The stated aim of the study was to quantify rates and risk factors for childhood unintentional injury in areas of rural China, where many children are left behind by migrant workers. The study analysed data from a cross-sectional survey of 3791 children. The paper is well structured and key information is present. Injury is leading cause of the death in the US and globally therefore the study topic is an important one, however as it stands the manuscript is not suitable for publication. The following is a summary of the key issues. I trust the authors will find this feedback useful.
Significant issues:
· Questionable ethics of getting children under the age of 15 to sign a consent form (line 97). Not sure this can be considered informed consent! Power imbalance of researchers and teachers cf. to a school child. In other parts of the manuscript it says primary caregivers gave consent (line 115)
· Accuracy of recall of a 9 year old
· Reliability of question items such as perceived wealth
· Unusual to discuss the intervention in the Discussion of this paper when the focus is on baseline data collection
Specific feedback for sections of the manuscript:
· Abstract:
o Aim is not consistent with the aim stated in the Introduction
· Introduction:
o Needs more extensive referencing for risk factors for child injury than one document. E.g. no mention of neighborhood factors
o Lines 46 – 49: need some context here as these leading causes will differ between high income and low income countries
o Line 76 – remove second ‘baseline’
· Methods:
o Insufficient information regarding risk factors examined and why they were selected.
o Why was no validation exercise undertaken e.g. survey parents/caregivers?
o Were any students excluded e.g. cognitive deficits?
· Results:
o No response rate given. How similar or dissimilar were those who took part compared with those who did not?
o Not convinced about the reliability of the family economic situation when children were asked to compare their wealth against others in the class –how do you then compare between schools???
o Table 1: age needs to note ‘in years’
o What about near drownings among those who could swim?
o Line 158: needs a close bracket after the first OR
· Discussion:
o Limitations need to highlight the reliability of risk factor recall in young children
o See earlier comment re discussing the intervention in this paper
General feedback:
· The title is not an accurate reflection of the content of the paper and should be revised
· Be consistent with the use of the LBC abbreviation e.g. line 73, line 80 etc.
Author Response
We very much appreciate the very useful comments of the reviewer.
We have answered point-by-point below in bold:
Significant issues:
· Questionable ethics of getting children under the age of 15 to sign a consent form (line 97). Not sure this can be considered informed consent! Power imbalance of researchers and teachers cf. to a school child. In other parts of the manuscript it says primary caregivers gave consent (line 115)
The purpose of the questionnaire was explained in detail, and our belief is that children have a good understanding of what they are undertaking. We have carried-out a large number of quantitative and qualitative studies in schoolchildren across the age range in China over many years, and have always been impressed by the openness and honesty of the children. Power imbalances between researchers and participants occur even among adults, especially vulnerable adults, and this is an inevitable consequence of research. In our view children participating in their own environments alongside classmates are actually less conscious of power imbalances. As we explained we also obtained permission from primary carers (See Line 116). This approach was approved by the Ethics Committees of University College London and Zhejiang University.
· Accuracy of recall of a 9 year old
We piloted the questionnaire partly in order to assess the accuracy of recall of younger children. We found it to be surprisingly good. Indeed, younger children seemed to demonstrate especially good recall, describing the trauma of the event in ways that older children did not. See Line 222
· Reliability of question items such as perceived wealth
Over many years of trying to find out the best ways of assessing children’s perception of household wealth in a Chinese setting, we have found this question about comparative wealth to be most useful, and a good measure of perceived inequality. It is this perceived inequality which we are most interested in, not absolute wealth. Indeed when we have presented this idea at meetings, many other Chinese researchers agree with us.
· Unusual to discuss the intervention in the Discussion of this paper when the focus is on baseline data collection
This may be unusual but it is totally deliberate. The point is to show that such research can lead to direct outcomes. Just collecting data, for the sake of it, strikes us as a wasted opportunity in this context of great need in many of these children. We actually think this is a major main strength of the paper.
Specific feedback for sections of the manuscript:
· Abstract:
Aim is not consistent with the aim stated in the Introduction
The Aim in the abstract is a shortened version of that in the introduction, because of the limited word count of the abstract. It is not intended as a verbatim copy, but covers the two most important points. To provide a shortened version of the aims in the abstract (because of the constraints of the word count) is normal practice, and indeed this approach is commonly taught.
· Introduction:
Needs more extensive referencing for risk factors for child injury than one document. E.g. no mention of neighborhood factors.
We have referenced a number of papers in this regard – all the papers cited from 1-4, 6,7, (and now the new 11 and 12) all include sections on risk factors. This includes neighbourhood factors which are often referred to as environmental factors in the more recent literature. We do mention important urban-rural differences in Line 46. However, we have cited two additional papers (citations 9 and 10 – see below) which do refer to neighbourhood specifically. We have adjusted all the references accordingly
Reading R, Langford A, Haynes R, Love A Accidents to preschool children: comparing family and neighbourhood risk factors Social Science & Medicine 1999:48(3):321-330
Burton P, Mulvaney C, Watson M. Relationships between child, family and neighbourhood characteristics and childhood injury: A cohort study Social Science & Medicine 2005; 61(9):1905-1915
Lines 46 – 49: need some context here as these leading causes will differ between high income and low income countries.
We have made additions here (Line 49) to clarify the context and differecnes between high and low income countries, although there are of course important overlaps between the two as well.
Line 76 – remove second ‘baseline’
Removed
Methods:
Insufficient information regarding risk factors examined and why they were selected.
As noted this study was part of a baseline survey informing an intervention to improve the psychosocial well-being of vulnerable children, especially left behind children. Therefore risk factors focused on the socio-demographic, left behind status, and primary carer. We think it is very unusual to be asked to justify the selection of standard risk factors in this way.
Why was no validation exercise undertaken e.g. survey parents/caregivers?
To validate the data in this setting would be very difficult to do, and would have limited value. First, the questionnaires were completed anonymously and we could not track individual parents or carers. Second, most parents are away and may not be aware of the occurrence of injury. Third, we know from spending time with these children that they are often afraid of telling grandparents about accidents, who are often frail, for fear of worrying them or having their freedom restricted. In our reading on this topic we have not found a paper which attempts formal validation.
Were any students excluded e.g. cognitive deficits?
No students were excluded specifically for such reasons. This would have been insensitive in the classroom setting. However, the total response rate was 94.1 %. Some of the returned or incomplete questionnaires may have been because of cognitive deficits, but we did not measure this specifically
· Results:
No response rate given. How similar or dissimilar were those who took part compared with those who did not?
Thank you for pointing-out our omission in this regard.
This questionnaire was administered in the classroom so the response rates were very high. We distributed 3992 questionnaires, of which 3876 (97.1%) were returned. Of these 85 were excluded from analysis because of failure to complete key variables, so the total response rate was 94.9% (n=3791). We believe that the most common reason for lack of completion was fear of asking for help from the Research Assistants, who were on hand. Because of limited completion we could not assess differences in left behind status between those who participated and those who did not. Even if we had known, given the small numbers involved, the differences would have been very marginal.
Not convinced about the reliability of the family economic situation when children were asked to compare their wealth against others in the class –how do you then compare between schools???
We agree that measurement of family economic situation is difficult in children when they don’t know what the family income is. Comparison between schools, villages, towns and counties could have been attempted, but this would have been a complicated data-dredging exercise which would not have contributed to the overall research questions. Our key finding was that left behind children felt better-off than those living with parents suggesting that parental migration at least leads to some financial benefits.
· Table 1: age needs to note ‘in years’
Done
· What about near drownings among those who could swim?
Of near drownings reported, none of the children could swim.
· Line 158: needs a close bracket after the first OR
Done
· Discussion:
· Limitations need to highlight the reliability of risk factor recall in young children
Done
· See earlier comment re discussing the intervention in this paper
Mentioned
General feedback:
· The title is not an accurate reflection of the content of the paper and should be revised
We assume the reviewer means the use of “environmental hazards” but this is not clear. If this is the case, we are happy to remove. Our preference is to leave it.
· Be consistent with the use of the LBC abbreviation e.g. line 73, line 80 etc.
Done
Reviewer 3 Report
Other than one minor typographical error Line 142 grammar and spelling excellent.
Line 91: Why this age group? Explain why <9yrs were not surveyed. I assumed this was because of not being school age or capable of completing a survey, but this should be elaborated.
Line 229: when were these interventions begun?
How long after the questionnaire?
Were there any preliminary results or are you doing further research?

Author Response
Many thanks for the constructive comments.
Other than one minor typographical error Line 142 grammar and spelling excellent.
Corrected
Line 91: Why this age group? Explain why <9yrs were not surveyed. I assumed this was because of not being school age or capable of completing a survey, but this should be elaborated.
Yes, this is because literacy in Chinese (being a character-based language) cannot be assumed until around the age of nine years. Even some of the over nine year olds need help with some characters.
Line 229: when were these interventions begun?
How long after the questionnaire?
Were there any preliminary results or are you doing further research?
We started to roll-out the intervention in a stepwise fashion across all five counties from March 2017.
The intervention has been ongoing for 18 months and we are currently analysing results. Preliminary results suggests that while knowledge about injury has improved, actual incidence of injury remains unchanged, but it is too early to draw firm conclusions.
Round 2
Reviewer 1 Report
I was interested in participation rate, that is relative to the total number enrolled, a smaller or larger proportion of whom may have been absent on survey days. However it may be that this information is hard to obtain. The query as to why there was such a high proportion of children cared for by neither parents nor grandparents (Table 1) has not been answered; if it is matter of failure to answer the question, why is the proportion in that column so much higher than in other columns?Author Response
It is now clear what the reviewer meant. We misunderstood.
This actually relates to a misunderstanding of the term primary carer, which is unambiguous in the Chinese setting, but clearly not everywhere. This refers to the adult who spends the most time parenting, ie carrying out parenting duties andactivities with the child. In this case, since the child is filling the questionnaire it a about the child's perception of who does the most parenting. So we show that ALL children left behind by both parents, (not surprisingly) see their grandparents as primary carers. But we also show that 22% of children living with both parents and 16% living with one parent see grandparents as primary carers, showing the importance of the grandparental caring role in Chinese society, largely because parents are the breadwinners.
To improve clarity, we have added a definition in Line 106.
"....the primary carer, that is the adult who takes most of the parenting role..."
Reviewer 2 Report
I do not agree with some of the responses to the issues I raised regarding the appropriateness of collecting data from young children, and the issues in relation to the SES measures used. The authors indicate this is consistent with other research from their country, which is of concern. But I feel they have addressed the issues I raised to an adequate level.
Author Response
We appreciate the reviewer's undertanding - and that most research in China has these limitations.